# Development of D-box peptides to inhibit the anaphase-promoting complex/cyclosome

Rohan Eapen[1], Cynthia Okoye[1], Christopher Stubbs[2], Marianne Schimpl[2], Thomas Tischer[3], Eileen J Fisher[2], Maria Zacharopoulou[1], Fernando Ferrer[4], David Barford[3], David R Spring[5], Catherine Lindon[1], Christopher Phillips[2], Laura S Itzhaki[1]*

[1]Department of Pharmacology, University of Cambridge, Cambridge, United Kingdom; [2]AstraZeneca, Cambridge, United Kingdom; [3]MRC Laboratory of Molecular Biology, Cambridge, United Kingdom; [4]p53 Laboratory, Singapore, Singapore; [5]Yusuf Hamid Department of Chemistry, University of Cambridge, Cambridge, United Kingdom

## eLife Assessment

The article represents a **fundamental** advance in designing peptide inhibitors targeting Cdc20, a key activator and substrate-recognition subunit of the APC/C ubiquitin ligase. Supported by **compelling** biophysical and cellular evidence, the study lays a strong foundation for future developments in degron-based therapeutics. The revised article has been strengthened by additional clarifications and data that address prior reviewer concerns. The work provides a robust framework for developing tools to manipulate protein degradation and will be of broad interest to researchers in protein engineering, cell cycle regulation, and targeted protein degradation.

*For correspondence:
lsi10@cam.ac.uk

**Abstract** E3 ubiquitin ligases engage their substrates via 'degrons' - short linear motifs typically located within intrinsically disordered regions of substrates. As these enzymes are large, multi-subunit complexes that generally lack natural small-molecule ligands and are difficult to inhibit via conventional means, alternative strategies are needed to target them in diseases, and peptide-based inhibitors derived from degrons represent a promising approach. Here we explore peptide inhibitors of Cdc20, a substrate-recognition subunit and activator of the E3 ubiquitin ligase the anaphase-promoting complex/cyclosome (APC/C) that is essential in mitosis and consequently of interest as an anti-cancer target. APC/C engages substrates via degrons that include the 'destruction box' (D-box) motif. We used a rational design approach to construct binders containing unnatural amino acids aimed at better filling a hydrophobic pocket that contributes to the D-box binding site on the surface of Cdc20. We confirmed binding by thermal-shift assays and surface plasmon resonance and determined the structures of a number of the Cdc20-peptide complexes. Using a cellular thermal shift assay, we confirmed that the D-box peptides also bind to and stabilise Cdc20 in the cell. We found that the D-box peptides inhibit ubiquitination activity of APC/C$^{Cdc20}$ and are more potent than the small-molecule inhibitor Apcin. Lastly, these peptides function as portable degrons capable of driving the degradation of a fused fluorescent protein. Interestingly, we find that although inhibitory activity of the peptides correlates with Cdc20-binding affinity, degradation efficacy does not, which may be due to the complex nature of APC/C regulation and effects of degron binding of subunit recruitment and conformational changes. Our study lays the groundwork for the further development of these peptides as molecular

therapeutics for blocking APC/C as well as potentially for harnessing the APC/C for targeted protein degradation.

## Introduction

E3 ubiquitin ligases recognise their substrates via short linear motifs (SLiMs) known as degrons on the substrates, which are often located in intrinsically disordered regions and bind with rather weak affinities (in the micromolar range) (*Tompa and Fuxreiter, 2008*; *Min et al., 2013*; *Guharoy et al., 2016*; *Hein and Nilsson, 2014*; *Pierce et al., 2016*). Recognition of the substrate by the proteasome depends on the number, type, and length of polyubiquitin chains and requires poly-ubiquitination through multiple rounds of recruitment of a ubiquitin-loaded conjugating E2 enzyme (UbcH10 or Ube2S) to the substrate-bound E3. There are numerous examples in which ubiquitination and degradation require the E3 to bind to multiple degrons on the substrate, a particularly striking case being the anaphase-promoting complex/cyclosome (APC/C), a giant multi-protein complex that controls progression of cells out of mitosis via the coordinated ubiquitination and degradation of over 100 different substrates (*Karamysheva et al., 2009*; *Di Fiore et al., 2016*; *Tian et al., 2012*; *Okoye et al., 2022*). APC/C function requires one of two coactivators, Cdc20 and Cdh1/FZR1 (*Min et al., 2013*; *Davey and Morgan, 2016*; *Bakos et al., 2018*), which comprise a substrate-binding WD40 domain and N- and C-terminal disordered regions that bind to other APC/C subunits and lead to conformational change and enhanced E2 binding. APC/C$^{Cdc20}$ activity is further controlled by the mitotic checkpoint complex (MCC) that inhibits APC/C in the presence of faulty chromosome attachments to the mitotic spindle (*Izawa and Pines, 2011*; *Di Fiore et al., 2016*; *Qiao et al., 2016*; *Alfieri et al., 2017*; *Watson et al., 2019*). APC/C$^{Cdc20}$ activity reduces as cells exit mitosis, the coactivator is switched to Cdh1/FZR1, and APC/C$^{FZR1}$ is active until the end of G1 phase. In addition to coactivator switching and post-translational modifications, the APC/C further controls the order of substrate degradation via differential degron-binding affinities (*Davey and Morgan, 2016*; *Alfieri et al., 2017*; *Bodrug et al., 2021*; *Okoye et al., 2022*), leading to different processivity of ubiquitination and chain linkages and thereby effective engagement of the proteasome, although the precise details of the relationship between APC/C-substrate interactions and the number, type, and length of polyubiquitin chains formed have not been fully resolved (*Figure 1*).

High-resolution structures of the APC/C and its complexes with substrates and E2s have provided tremendous insights into the mechanism of ubiquitination and degradation (*Barford, 2020*). Coactivators Cdc20 and FZR1 have at least three binding sites for substrate degrons: the 'Destruction-box' (D-box), the KEN motif, and the ABBA motif (thought to be required for Cyclin A degradation only; *Qin et al., 2017*). A study of the structure of APC/C-FZR1 in complex with its pseudo-substrate inhibitor Acm1 showed that all three of the Acm1 degrons can bind simultaneously (*He et al., 2013*). The KEN motif is recognised by the 'top' surface of the WD40 beta-propeller domain of the co-activator, and the ABBA motif binds to the 'side' of the beta-propeller at one of its blades. The D-box binding site comprises a cleft between two of the beta-propeller blades and the neighbouring APC/C subunit APC10, which is thought to result in stabilisation of the active complex (*Burton and Tsakraklides, 2005*; *Buschhorn et al., 2011*; *da Fonseca et al., 2011*; *Chang et al., 2014*; *Matyskiela and Morgan, 2009*; *Qin et al., 2019*). Most recently, single-molecule studies have shed new insights into the key role of degron multivalency in enabling efficient substrate ubiquitination and degradation (*Hartooni et al., 2022*).

Inhibitors of APC/C$^{Cdc20}$ activity represent an interesting therapeutic approach to target dividing cells in cancer. Given the large size of the APC/C machine (14 subunits) and the complex mechanisms described above that regulate its function, it is not surprising that it is challenging to inhibit. Apcin and TAME are recently identified small-molecule inhibitors, but they have limited activity and complicated output (*Richeson et al., 2020*; *Sackton et al., 2014*). In this paper, we use a rational design approach, based on D-box consensus sequences and a 'Super D-box' peptide derived from Hsl1, and examination of the Cdc20-degron interface, to design a series of more potent binders containing unnatural amino acids aimed at better filling the D-box hydrophobic pocket on the interaction interface. We quantified binding by thermal shift assays (TSAs) and surface plasmon resonance (SPR) and used a cellular thermal shift assay (CETSA) to demonstrate target engagement within the cellular context. The peptides also show functional engagement with APC/C in the cell as evidenced by their ability

to drive the degradation of a fluorescent protein. Most strikingly, in vitro ubiquitination assays with recombinant APC/C$^{Cdc20}$ show that these peptides are more potent inhibitors of Cyclin B1 ubiquitination than Apcin. Interestingly, we find that although inhibitory activity of the peptides correlates with Cdc20-binding affinity, their degradation efficacy does not. This may be due to the complex nature of APC/C degrons and their bipartite interaction with different subunits, role in E2 recruitment, and consequent impact of positioning for effective ubiquitination. The results provide a useful starting point for the further development of these peptides as molecular therapeutics for blocking APC/C, as well as potentially also for harnessing APC/C for targeted protein degradation.

## Materials and methods

### Cloning, expression, and purification of Cdc20

DNA encoding residues 161–477 of human Cdc20 (Cdc20$^{WD40}$) was cloned into a pU1 vector with an N-terminal His$_6$-tag followed by a TEV protease cleavage site. The plasmid was then transformed into DH10 MultiBac cells expressing a Cre-recombinase. Positive clones were grown up and bacmid DNA prepared by standard protocols. Sf21 or Sf9 cells were grown at 27°C in Erlenmeyer flasks (Corning) and maintained in mid-log phase of growth prior to all experiments. High-titre baculovirus was produced by transfecting bacmid DNA into Sf21 cells at $0.5 \times 10^6$ cells/ml cells using Superfect (QIAGEN) in 24 deep-well blocks. Virus was harvested 1 week post transfection. For protein over-expression, Sf21 or Sf9 cells were infected with the virus stock and harvested about 60 hours post infection. Cell pellets were resuspended in 50 mM Tris-HCl, 300 mM NaCl, 1 mM MgCl$_2$, 1 mM TCEP, 5% (v/v) glycerol, SigmaFAST EDTA-free protease inhibitor cocktail (1 tablet/100 ml), pH 8.5. Resuspended pellets were lysed by one freeze–thaw cycle at –80°C. Lysates were then clarified by centrifugation at 45,000×$g$ for 45 minutes at 4°C. Supernatants were flowed over a 5 ml HisTrap

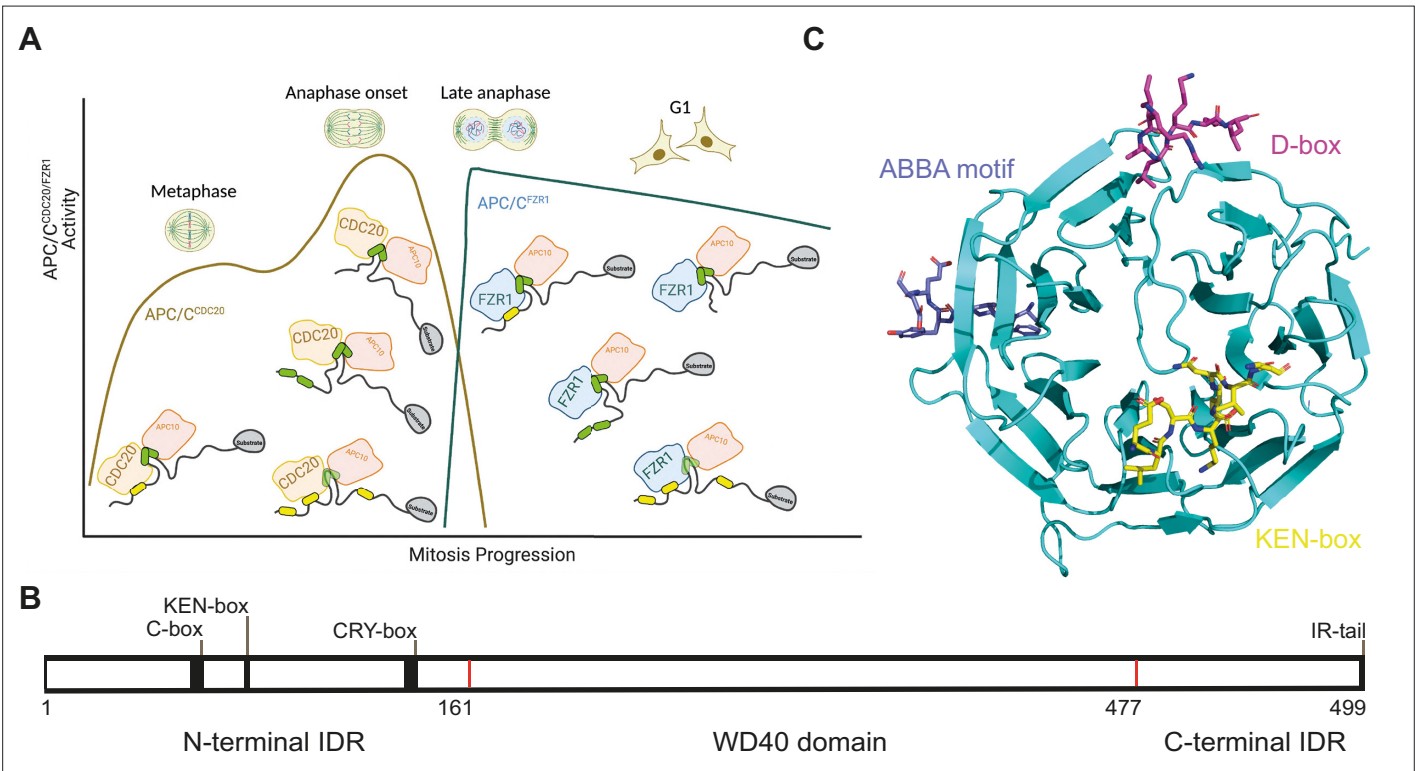

**Figure 1.** Structure and function of anaphase-promoting complex/cyclosome (APC/C). (**A**) Schematic of APC/C activity during mitotic exit, indicating the switch in co-activator from Cdc20 to FZR1. Most substrates contain variable degrons (D-box in green, KEN in yellow) present in IDRs. (**B**) Domain structuring of Cdc20 comprising an N-terminal IDR with the C-box, KEN-box, and CRY-box motifs, the central WD40 domain responsible for substrate recruitment via the degron binding sites and the C-terminal IDR containing the IR-tail. (**C**) Schematic of the structure of the Cdc20 WD40 domain (PDB: 4GGC) overlaid with those of the WD40 domain in complex with Acm1 D-box and ABBA motif peptides (PDB: 4BH6) (*He et al., 2013*) and the KEN-box peptide 4GGD (*Tian et al., 2012*).

Excel column and washed with 20 column volumes (CV) of 50 mM Tris-HCl, 300 mM NaCl, 10 mM imidazole, 1 mM $MgCl_2$, 1 mM TCEP, 5% (v/v) glycerol, pH 8.5. Proteins were then eluted with the above buffer including 300 mM imidazole directly into a 26/10 desalting column pre-equilibrated in the above buffer without imidazole. Eluted protein fractions were then pooled, and the $His_6$-tag was removed using $His_6$-TEV protease (S219V) overnight at 4°C. Proteins were then flowed back over a 5 ml HisTrap Excel column, collecting the flow-through containing the non-tagged Cdc20 protein. Protein eluent was then diluted in 25 mM Tris-HCl buffer, 1 mM $MgCl_2$, 1 mM TCEP, 5% (v/v) glycerol, pH 8.5 to a final concentration of 30 mM NaCl. Proteins were then loaded onto a MonoQ 10/100 GL column and eluted over 20 CV with 1 M NaCl. Protein fractions containing the Cdc20 protein were then pooled and concentrated before separating on a Superdex 75 increase 10/300 GL column in the final buffer containing 25 mM Tris-HCl, 150 mM NaCl, 1 mM $MgCl_2$, 5 mM TCEP, 5% (v/v) glycerol, pH 8.5.

## Minimal biotinylation of Cdc20 for SPR

The protocol for minimal biotinylation was adapted from (*Papalia and Myszka, 2010*). Purified $Cdc20^{WD40}$ was diluted into the reaction buffer (10 mM Tris-HCl, 150 mM NaCl, 1 mM $MgCl_2$, 5% [v/v] glycerol, 1 mM TCEP, pH 7.3). A 0.9:1 molar ratio of Sulfo-NHS-LC-LC-Biotin (Thermo Fisher Scientific, A35358) was added to the diluted Cdc20 protein. The contents were briefly mixed by vortex and incubated on ice for 3 hours. The sample was then separated on a Superdex 75 Increase 10/300 GL column to remove free biotin.

## Peptide synthesis and purification

Peptide synthesis was performed on a 0.1 mmol scale using Ramage-ChemMatrix resin (Sigma-Aldrich). Fmoc-L-amino acids (2 eq.), HATU (2 eq.), and HOAt (2 eq.) were dissolved in 2 ml of NMP. DIEA (3.4 eq.) was used to activate the coupling mixture. Activated Fmoc-L-amino acids were coupled for 10 minutes (Fmoc-L-Arginine, 2×5 eq., 30 minutes). Resins were washed in DMF and deprotected in 20% piperidine in DMF for 15 minutes. All peptides were N-terminally acetylated in 4 ml DMF, 4 ml acetic anhydride, 2 mL DIEA for 10 minutes. A peptide cleavage cocktail consisting of 93% TFA, 3.5% TIPS, and 3.5% $ddH_2O$ was used to deprotect and cleave the peptide from the resin for 1 hour. The eluate was triturated by the addition of diethyl ether, and the resulting precipitate was isolated by brief centrifugation. All peptides were characterised by LCMS using a Waters ACQUITY H-Class UPLC with an ESCi Multi-Mode Ionisation Waters SQ Detector 2 spectrometer. LC was performed on a ACQUITY UPLC CSH C18 (2.1 mm × 50 mm, 1.7 µm, 130 Å) at 40°C, with a PDA e $\lambda$ detector 220–800 nm, interval 1.2 nm. The following solvents and gradients were used for LC runs. Solvent A: 2 mM $NH_4OAc$ in 95% $H_2O$, 5% MeCN, solvent B: 100% MeCN, solvent C: 2% formic acid from 5 to 95% B with a constant of 5% C over 1 minute at 0.6 ml/min. Analytical and semi-preparative HPLC runs were performed on an Agilent 1260 Infinity system using a Supelcosil ABZ +PLUS (150 mm × 4.6 mm, 3 µm) and Supelcosil ABZ +PLUS (250 mm × 21.2 mm, 5 µm), respectively. Peptides were eluted with a linear gradient system (solvent A: 0.1% TFA in $H_2O$, solvent B: 0.05% TFA in MeCN) over 15 minutes at 1 ml/min and 20 minutes at 20 ml/min, respectively. Eluents were monitored by UV absorbance at 220 nm and 254 nm. Analytical data for all peptides are shown in *Figure 3—figure supplements 2–4* and *Figure 4—figure supplements 1 and 2*.

## Thermal shift assays

Assays were performed using a Roche LightCycler 480 I in 96-well plate format. Each well (20 µl) was prepared with 750 nM of purified $Cdc20^{WD40}$ and varying concentrations of D-box peptides, Apcin or DMSO (vehicle control) in assay buffer; 25 mM Tris-HCl, 150 mM NaCl, 1 mM $MgCl_2$, 5% (v/v) glycerol, 1 mM TCEP, 1% (v/v) DMSO, 5× SYPRO Orange (Thermo Fisher), pH 8.5. Thermal ramps were conducted from 25°C to 95°C at a rate of 0.03°C/second, and data were collected at a frequency of 20 points/°C. An excitation wavelength of 483±35 nm was used to excite SYPRO Orange, and the fluorescence emission was detected at a 568±20 nm. Measurements were performed in triplicate and errors listed are the standard deviation. Melting temperatures were determined by the minima peak of the negative differential in the 'Tm calling' analysis within the in-built analysis software.

## Surface plasmon resonance assays

Experiments were performed using a Biacore T200 instrument (GE Healthcare) at 15°C. Biotinylated-Cdc20$^{WD40}$ was immobilised onto a SA biosensor chip (GE Healthcare) in running buffer (10 mM HEPES, 150 mM NaCl, 0.1 mM TCEP, 0.05% [v/v] Tween 20 and 1% [v/v] DMSO, pH 7.4) overflow cells 2, 3, and 4 at varying ligand densities. Flow cell 1 was used as a reference cell. Free biotin binding sites were blocked using amine-PEG$_4$-Biotin. Peptides and Apcin analytes were diluted from DMSO stock solutions in running buffer without DMSO and were buffer matched to 1% DMSO. Titrations of each analyte were run over the sensor chip at a flow rate of 30 µl/min. Binding interactions were detected as a change in response units over the reference flow cell and subtracted from a blank buffer injection. Dissociation constants ($K_D$) were calculated by fitting the response units (RUs) at steady-state equilibrium generated by the binding of an analyte to Cdc20$^{WD40}$ against the concentration of analyte using the following equation:

$$RU_{\text{analyte}} = \frac{RU_{\max} \times [\text{analyte}]}{\left(K_D + [\text{analyte}]\right)}$$

where $RU_{\text{analyte}}$ is the response units at equilibrium during a given injection of a concentration of analyte, [analyte]. $RU_{\max}$ is the maximum response produced by the analyte, dependent on the RU of immobilised ligand on a given flow cell. $K_D$ is the dissociation constant of a given analyte to the ligand. $K_D$ values are shown as the average of measurements from the three reference-subtracted flow cells.

## Cellular thermal shift assays

Full-length Cdc20 (residues 1–499) with a C-terminal HiBiT tag (GS**VSGWRLFKKIS**GS, Promega) was cloned into a pcDNA3.1(-) vector. HEK 293T cells were cultured in DMEM + 10% FBS (Sigma-Aldrich, F7524) at 5% CO$_2$ in a humidified environment. Cells were grown to 70% confluency in T75 flasks prior to transient transfection with 10 µg of Cdc20_HiBiT_pcDNA3.1(-) plasmid with Lipofectamine 2000 (Invitrogen, Thermo Fisher Scientific) according to the manufacturers' protocol. Cells were harvested after 48 hours by trypsinisation and were subsequently washed twice in PBS with repeated centrifugation at 1000×$g$ for 2 minutes. The pellet was then resuspended in lysis buffer (PBS, 1×Sigma-FAST EDTA-free protease inhibitor tablet [Sigma-Aldrich], 2 mM NaVO$_3$, 5 mM NaF, pH 7.4) and freeze–thaw lysed in liquid nitrogen. The lysate was clarified by centrifugation at 20,000×$g$, 4°C for 20 minutes and the protein concentration of the supernatant was quantified by BCA (Pierce). Lysates were used at a final concentration of 0.2 mg/ml in lysis buffer. Lysates were aliquoted in 300 µl and were spiked with D-box binding site ligands to a concentration of 100 µM maintaining 1% DMSO. Compounds were incubated on ice for 30 minutes prior to aliquoting further into PCR strip tubes on a PCR block at 4°C. Lysate aliquots were then heated on a second PCR block at the indicated temperatures for 3 minutes prior to returning to 4°C. Heated lysates (5 µl) were then transferred into an AlphaPlate light-grey 384-well plate in quadruplicate by multichannel pipette. Nano-Glo HiBiT lytic detection system (Promega) was diluted as per the manufacturer's instructions and 5 µl were added to each well by multichannel pipette. Lysis buffer and a non-transfected HEK 293T cell lysate were used as negative controls. Following 5 minutes of incubation on a plate shaker, the plate was measured using a CLARIOStar microplate reader (BMG Labtech), with the detector set to read at 460±80 nm, the focal height at 10.5 cm, and the gain adjusted to 2000. Data were normalised to the unheated sample (4°C) and were fitted using a Boltzmann equation to extract the melting temperature (T$_m$) (*Niesen et al., 2007*).

## Protein crystallisation

Peptide D21 was added to Cdc20$^{WD40}$ in a stoichiometric manner and was co-concentrated to 1.9 mg/ml. The resulting complex was crystallised in a 2:1 protein to well solution ratio at 20°C using the sitting-drop vapour-diffusion method with a well solution containing 0.1 M MES pH 6.5, 12% (w/v) PEG 6000, 10% (v/v) MPD for Cdc20$^{WD40}$-D21 and 0.1 M MES pH 6.5, 14% (w/v) PEG 6000, 10% (v/v) MPD for Cdc20$^{WD40}$-D20 and Cdc20$^{WD40}$-D7. Crystals grew to a maximum size after 3 days of incubation. For soaking experiments, crystals were first looped and washed through three drops containing 0.1 M MES pH 6.5, 20% (w/v) PEG 6000 to wash out MPD from the crystal. Crystals were then looped and incubated in a solution containing 0.1 M MES pH 6.5, 20% (w/v) PEG 6000 and 2.5 mM D21 or

D20 (5% (v/v) DMSO) or D7 (10% [v/v] DMSO) for 4 hours. Soaked crystals were cryo-protected in the soak solution supplemented with 10% (v/v) glycerol and were flash frozen in liquid nitrogen.

### Data collection and structure determination

Diffraction data were collected on beamline I04 at the Diamond Light Source (Oxford, UK) and processed using autoPROC-STARANISO STARANISO (*Vonrhein et al., 2018*). Phases were obtained by molecular replacement using the crystal structure of human Cdc20 (PDB ID code 4GGC) as the search model (*Tian et al., 2012*). Iterative model building and refinements were performed with COOT and BUSTER, respectively (*Emsley et al., 2010*, *Bricogne et al., 2020*). Cdc20-D20 and Cdc20-D21 datasets were first refined using Refmac5 within the CCP4i suite (*Winn et al., 2011*; *Kovalevskiy et al., 2018*) before final refinements using BUSTER. Data collection and structure refinement statistics are summarised in *Supplementary file 1A*.

### Ubiquitination assays

In vitro ubiquitination experiments were performed using APC/C and Cdc20 purified from insect cells (*Zhang et al., 2016*). 60 nM APC/C, 30 nM Cdc20, 90 nM UBA1, 300 nM UbcH10, 300 nM Ube2S, 35 mM ubiquitin, 1 mM cyclin B1, 5 mM ATP, 10 mM $MgCl_2$, were mixed in a buffer containing 40 mM HEPES (pH 8.0), 80 mM NaCl, 0.6 mM DTT. The reaction was either performed with the indicated concentrations of peptides or DMSO (Sigma-Aldrich) as the vehicle control. The reaction was incubated for 30 minutes at 23°C and stopped by the addition of one volume of 2× concentrated NuPAGE LDS loading buffer (Invitrogen).

### Protein degradation assays

The pEGFP-N1 vector was modified by swapping the EGFP-coding sequence for mNeon-coding sequence using the *AgeI/NotI* cloning sites. The Aurora kinase A (AURKA) C-terminal fragment (364-403) containing the non-degron $R_{371}xxL$ motif (D0) together with an extended IDR was amplified by PCR and cloned into the modified vector with *BamHI/AgeI* sites. Round the horn site-directed mutagenesis was used to generate different D-box variants and validated by DNA sequencing. U2OS cells were cultured in DMEM supplemented with 10% FBS, 200 µM Glutamax-1, 100 U/ml penicillin, 100 µg/ml streptomycin, and 250 ng/ml fungizone (all from Thermo Fisher Scientific) at 37°C in humidified atmosphere containing 5% $CO_2$. Plasmids were introduced into U2OS cells by electroporation using the Neon Transfection System 10 µl Kit (Thermo Fisher Scientific) and cells seeded on eight-well microscopy slides (Ibidi) and recovered for 24 hours. DMEM medium was exchanged for phenol red-free Leibovitz's L15 (Thermo Fisher Scientific), supplemented as above. Time-lapse imaging was conducted at 37°C using a widefield imaging platform composed of Olympus IX83 motorised inverted microscope, Spectra-X multi-channel LED widefield illuminator (Lumencor, Beaverton, OR, USA), Optospin filter wheel (Cairn Research, Faversham, UK), CoolSnap MYO CCD camera (Photometrics, Tucson, AZ, USA), automated XY stage (ASI, Eugene, OR, USA), and climate chamber (Digital Pixel, Brighton, UK), all controlled using Micro-Manager software (*Edelstein et al., 2014*). Fluorescence and phase contrast images of cells in mitosis were acquired with a ×40 objective binned at 2 × 2 at 2-minute intervals. Fluorescence intensity of mNeon in individual mitotic cells was quantified from 16-bit tiff files using ImageJ by integrating pixel measurements after subtraction of background fluorescence. Degradation curves were synchronised in silico to anaphase onset to generate average curves for multiple cells in each experimental condition.

## Results

### Quantification of Cdc20-binding activity of the small-molecule Apcin

We first produced Cdc20 protein in sufficient quantities for biophysical analysis and then used the known small molecule binder, Apcin, to test that the purified protein was functional and to benchmark our peptide-binding measurements. As Cdc20 comprises a WD40 domain that binds to the different degrons and is flanked on each end by long intrinsically disordered regions, we made a construct comprising the WD40 domain (residues 161–477) with an N-terminal $His_6$-tag and expressed this protein in baculovirus as previously described (*Sackton et al., 2014*; *Tian et al., 2012*). We biotinylated Cdc20 at a single site, as shown by electrospray-ionisation mass spectrometry for

streptavidin-mediated ligand capture in SPR (*Figure 2—figure supplement 1*). Using TSA and SPR, we confirmed that the purified Cdc20 was capable of binding to Apcin. The $K_D$ obtained by SPR was 420±50 nM (*Figure 2*).

## Design of D-box peptides

We focused on D-box peptides, as there is much evidence from the literature that points to the unique importance of the D-box motif in mediating productive interactions of substrates with the APC/C (i.e. those leading to polyubiquitination & degradation). One of the clearest examples is a study that tested the degradation of 15 substrates of yeast APC/C in strains carrying alleles of Cdh1 in which the docking sites for D-box, KEN, or ABBA were mutated (*Qin et al., 2017*). The

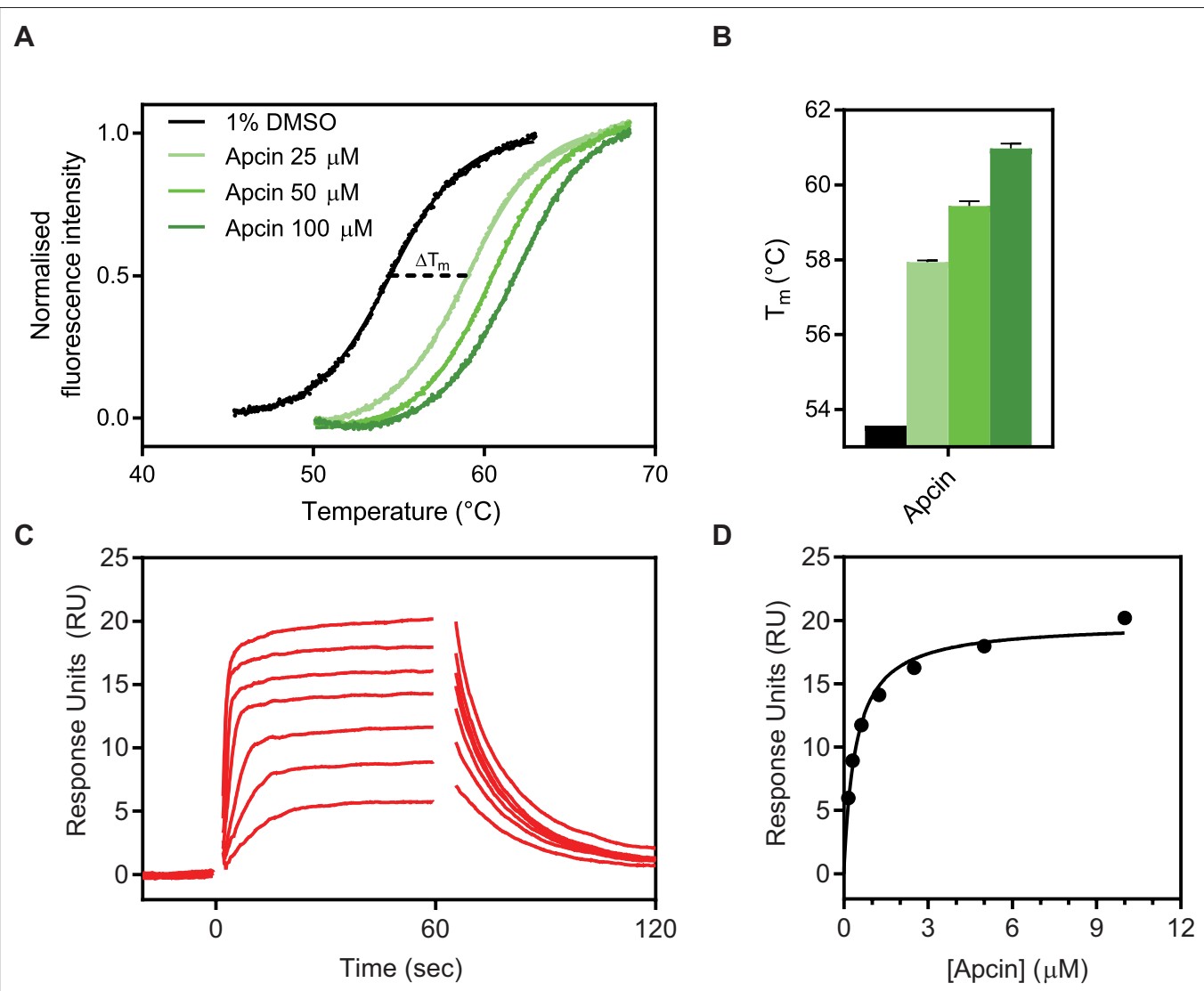

**Figure 2.** Biophysical characterisation of Apcin binding to Cdc20^WD40 by thermal shift assay (TSA) and surface plasmon resonance (SPR). (**A**) Representative examples of thermal unfolding traces of Cdc20 ^WD40 in the presence of 1% DMSO as the vehicle control or Apcin at concentrations of 25, 50, and 100 μM. (**B**) Corresponding melting temperatures calculated from derivative plots of the thermal unfolding traces. Mean data from triplicate measurements are shown, with error bars representing standard deviations. (**C**) Reference-subtracted sensorgrams of biotinylated Cdc20^WD40 and Apcin. (**D**) Binding affinity determination of Apcin to Cdc20^WD40 domain by steady-state analysis of the sensorgrams.

The online version of this article includes the following source data and figure supplement(s) for figure 2:

**Source data 1.** TSA data and BLI sensorgram data for *Figure 2*.

**Figure supplement 1.** Verification of single-site biotinylation of purified Cdc20 WD40 domain (residues 161-477) using Sulfo-NHS-LC-LC-Biotin (Thermo Scientific, A35358).

authors observed that, whereas degradation of all 15 substrates depended on D-box binding, only a subset required the KEN binding site on Cdh1 and only one required the ABBA binding site. A more recent study (*Hartooni et al., 2022*) of binding affinities of different degron peptides concluded that KEN motif has very low affinity for Cdc20 and is unlikely to mediate degradation of APC/C-Cdc20 substrates. Engagement of substrate with the D-box receptor is therefore the most critical event mediating APC/C activity and the interaction that needs to be blocked for most effective inhibition of substrate degradation.

Crystal structures of the D-box-coactivator interactions (*Chao et al., 2012*; *He et al., 2013*) show that there are three key residues, Arginine 1, Leucine 4, and Asparagine 9 in the RxxLxxxxN D-box motif (*Figure 3A*). Leucine 4 contacts a hydrophobic pocket in the co-activator subunit, whereas the 'tail' of the D-box degron and its flanking sequence (residues 8–12) contact the APC10 subunit. As a starting point we used two peptides, a 10-residue consensus-like sequence derived from Hsl1 (**D1: GRAALSDITN**) (*Burton and Tsakraklides, 2005*; *Frye et al., 2013*; *Davey and Morgan, 2016*), and a 9-residue consensus sequence based on known D-box degrons from APC/C substrates (**D2: RLPLG-DISN**) (*He et al., 2013*). TSA and SPR showed that **D1** binds to Cdc20$^{WD40}$ with a weak affinity ($K_D$ = 18.6 ± 0.2 μM) (*Table 1*). **D2** had no detectable affinity by TSA and, consequently, was not analysed by SPR. We hypothesised that the apparent lack of **D2** binding may be due to its low solubility in aqueous buffer rather than an inability to bind. Based on the Cdh1-Acm1 structure, the side chain of the amino acid at position 2 is likely to be solvent exposed in the context of Cdc20$^{WD40}$ (*Figure 3*). Given that we observed a measurable affinity with D1, we chose to incorporate Ala (instead of Leu) at position 2 in all subsequent peptides.

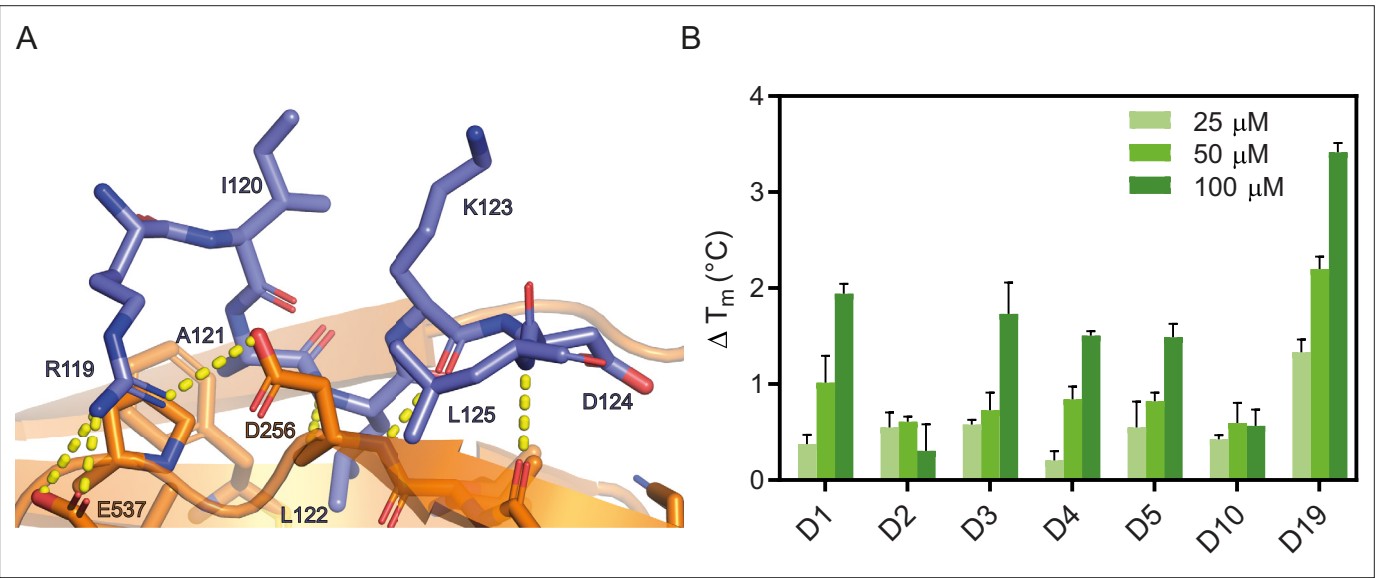

**Figure 3.** D-box peptide mutations. (**A**) Schematic showing the Acm1 D-box peptide bound to yeast FZR1 homologue Cdh1. R119 of the D-box forms H-bond interactions with D256 and E537 of Cdh1. L122 of the D-box buries into the canonical pocket on the surface of Cdh1 (PDB: 4BH6, *He et al., 2013*). (**B**) Melting temperature of Cdc20$^{WD40}$ in the presence of D-box peptides at 25, 50, and 100 μM concentrations, calculated from derivative plots of the thermal unfolding traces. Mean data from triplicate measurements are shown, with error bars representing standard deviations.

The online version of this article includes the following source data and figure supplement(s) for figure 3:

**Source data 1.** TSA melting temperature data for *Figure 3B*.

**Figure supplement 1.** Reference-subtracted surface plasmon resonance (SPR) sensorgrams and binding curves for (**A**) D1, (**B**) D3, (**C**) D4, (**D**) D5, (**E**) D10, (**F**) D19 binding to Cdc20.

**Figure supplement 2.** Linear amino acid sequence and chemical structures for peptides D1 and D2 are illustrated alongside the expected exact mass and molecular weights for each peptide.

**Figure supplement 3.** Linear amino acid sequence and chemical structures for peptides D3 and D4 are illustrated alongside the expected exact mass and molecular weights for each peptide.

**Figure supplement 4.** Linear amino acid sequence and chemical structures for peptides D5, D10 and D19 are illustrated alongside the expected exact mass and molecular weights for each peptide.

**Table 1.** Binding of D-box peptides to Cdc20$^{WD40}$ measured by surface plasmon resonance and thermal shift assay.

| Peptide | Sequence | $K_D$ (µM) | $\Delta T_m$ (°C at 100 µM peptide) |
|---|---|---|---|
| D1 | Ac-GRAALSDITN-NH$_2$ | 18.6±0.2 | 1.9±0.1 |
| D2 | Ac-RLPLGDISN-NH$_2$ | n.d. | 0.3±0.3 |
| D3 | Ac-RAPLGD**V**SN-NH$_2$ | 54.4±0.7 | 1.7±0.3 |
| D4 | Ac-RAPLGD**I**SN-NH$_2$ | 19.6±0.2 | 1.51±0.05 |
| D5 | Ac-RAPLGD**L**SN-NH$_2$ | 27±1 | 1.5±0.1 |
| D10 | Ac-RA**A**LGDISN-NH$_2$ | 70±3 | 0.6±0.2 |
| D19 | Ac-RA**P**LSDITN-NH$_2$ | 5.9±0.1 | 3.4±0.1 |

n.d. indicates not detectable.

## Isoleucine at position 7 and proline at position 3 of the D-box peptide are optimal for binding

From the consensus sequence, we observed that substrate proteins have approximately equal frequency of Val, Leu, and Ile at position 7.Figure 3A Given the similar structural and physical properties of the three aliphatic side chains, we compared peptides with each of these three amino acids at position 7 and found that **D4** with Ile7 had the highest affinity for Cdc20 (1.5-fold higher than **D5** with Leu7; 19.6±0.2 µM and 27±1 µM respectively) (*Table 1* and *Figure 3—figure supplement 1C and D*). Interestingly, the shorter hydrocarbon chain of Val in **D3** gave the weakest affinity, with a $K_D$ determined by SPR at 54.4±0.7 µM (*Figure 3—figure supplement 1B*).

We next investigated the contribution of proline versus alanine at position 3 (*Table 1*). Like the position 7 residues, Pro and Ala appear in approximately equal distribution to each other among known substrate proteins. In the context of D-box degron binding, modelling of our **D4** peptide to the *Saccharomyces cerevisiae* Cdh1 structure showed that Pro3 may form a favourable turn in the D-box peptide backbone to allow the side chain of Leu 4 to adopt its canonical pocket (*Figure 3A*). We also note that relative to alanine, the side chain of Pro3 forms additional van der Waals contacts with Tyr207 of Cdc20 (Tyr210 of Cdh1). To test this hypothesis, we synthesised **D10**, a derivative of **D4** containing a P3A single point mutation. As expected, this mutation was significantly detrimental with an affinity of 70±3 µM by SPR and in parallel a loss of thermal stabilisation by TSA (*Figure 3B*, *Figure 3—figure supplement 1E*). Upon confirming our hypothesis, we synthesised a derivative of **D1** containing the A3P point mutation, yielding **D19** (**RAPLSDITN**). This substitution resulted in a three-fold increase in affinity ($K_D$ = 5.9 ± 0.1 µM) compared with its parental sequence (*Figure 3—figure supplement 1F*, *Table 1*).

**Table 2.** Binding of D-box peptides containing unnatural amino acids replacing Leu4 binding to Cdc20$^{WD40}$ measured by surface plasmon resonance and thermal shift assay.
Reported values are the mean ± standard deviation of triplicate measurements.

| Peptide | Sequence | $K_D$ (µM) | $\Delta T_m$ (°C, at 100 µM peptide) |
|---|---|---|---|
| D7 | Ac-RAP**C**$_3$GDISN-NH$_2$ | 3.1±0.1 | 4.13±0.03 |
| D12 | Ac-RAA**C**$_3$GDISN-NH$_2$ | 13.3±0.1 | 2.0±0.3 |
| D20 | Ac-RAP**C**$_3$SDITN-NH$_2$ | 0.90±0.01 | 6.3±0.1* |
| D21 | Ac-RAP**F**$_3$SDITN-NH$_2$ | 0.52±0.01 | 6.7±0.1 |

*Standard deviation could not be determined, and the error was estimated based on uncertainties in peptide and protein concentrations.

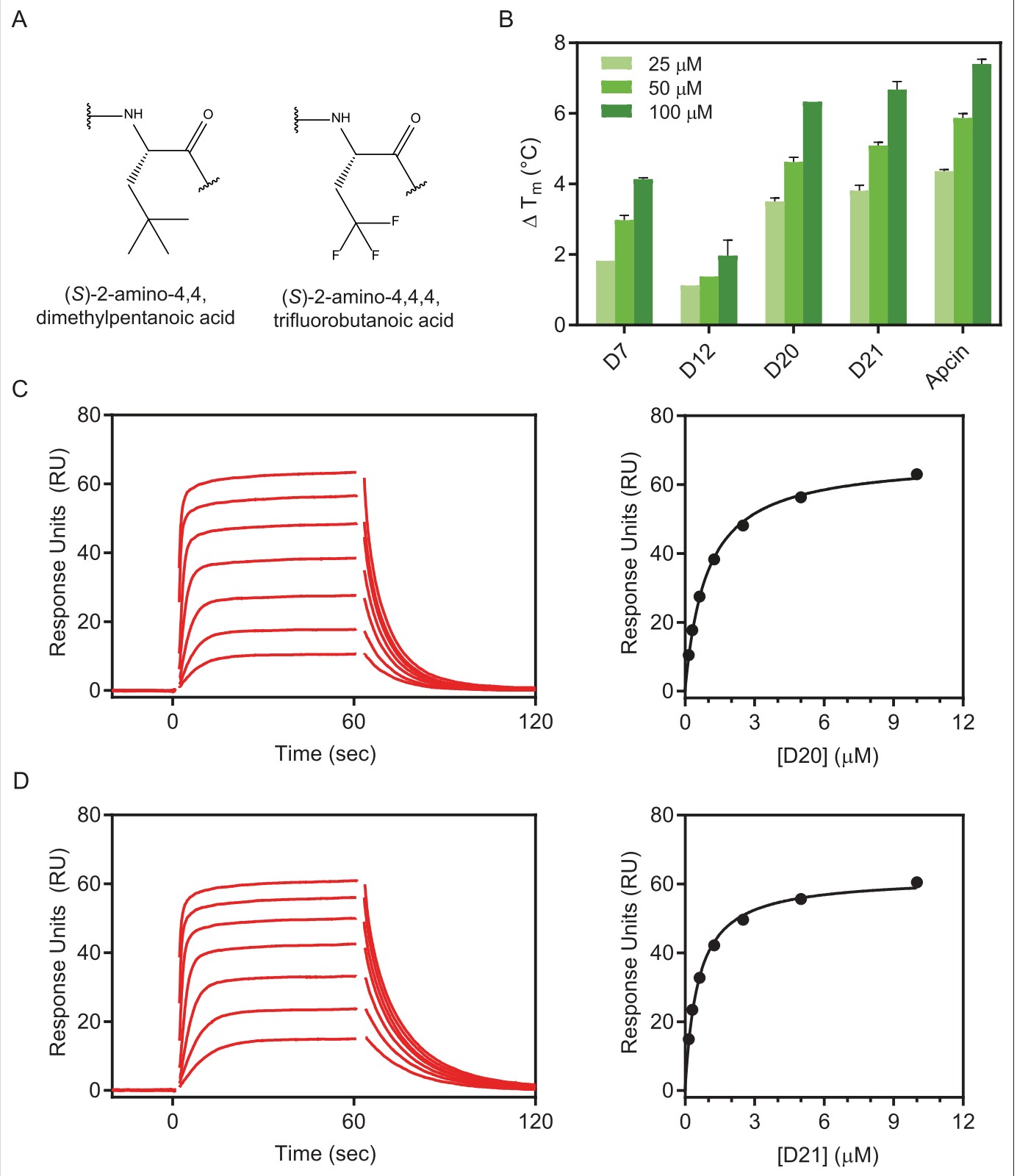

**Figure 4.** D-box peptides incorporating unnatural amino acids. (**A**) Schematics of the two unnatural amino acids used. (**B**) Thermal stabilisation of the Cdc20$^{WD40}$ by the two highest affinity peptides **D20** and **D21** calculated from derivative plots in thermal shift assay (TSA) . Surface plasmon resonance (SPR) reference-subtracted sensorgrams and binding curves for (**C**) **D20** and (**D**) **D21**.

The online version of this article includes the following source data and figure supplement(s) for figure 4:

*Figure 4 continued on next page*

*Figure 4 continued*

**Source data 1.** BLI sensorgram data and TSA melting temperature data for *Figure 4*.

**Figure supplement 1.** Linear amino acid sequence and chemical structures for peptides D7 and D12 are illustrated alongside the expected exact mass and molecular weights for each peptide.

**Figure supplement 2.** Linear amino acid sequence and chemical structures for peptides D20 and D21 are illustrated alongside the expected exact mass and molecular weights for each peptide.

## Unnatural amino acids at position 4 of the D-box peptide result in significantly enhanced binding affinity to Cdc20

The surface topology of Cdc20 is largely flat, making it hard to drug. Nevertheless, in Apcin the tri-chlorinated moiety makes particular use of the Leu4-binding pocket on Cdc20. Taking inspiration from the small molecule, we explored candidate unnatural amino acids to incorporate into the D-box peptides at position 4. Given that the pocket can accommodate a tri-chlorinated carbon moiety within Apcin, we explored similar moieties to append to our D-box peptides. We incorporated (*S*)–2-amino-4,4-dimethylpentanoic acid (**C₃**) (*Figure 4A*) into the backbone sequences of **D4**, **D10,** and **D19** replacing leucine at position 4, yielding peptides **D7**, **D12**, and **D20**, respectively (*Table 2*). As expected, the structure–activity relationship (SAR) held true between all peptides, whereby incorporation of the unnatural amino acid increased the binding affinity over 6-fold versus the respective parental peptide (*Table 2*). Building on this success, we further explored the commercially available halogenated analog, (*S*)–2-amino-4,4,4-trifluorobutanoic acid (**F₃**) (*Figure 4A*), leading to peptide **D21** (*Table 2*). With the tri-fluorinated group, a further increase in binding affinity was achieved ($K_D = 520 \pm 10$ nM), which is similar to that of Apcin.

## Crystal structures of Cdc20-peptide complexes reveal D-box binding mode

Previous attempts to co-crystallise Cdc20 and securin-derived or cyclin B1-derived D-box peptides by Tian and co-workers were unsuccessful (*Tian et al., 2012*), which may be due to the low affinity of peptides comprising these sequences. Despite the relatively high affinity of **D21** and the approximate 1:1.5 ratio of protein to peptide used in co-crystallisation experiments, crystals were absent of peptide ligands and instead contained the 2-methyl-2,4-pentanediol (MPD) molecule in the leucine-(P4) binding cleft (data not shown), originating from the crystallisation solution. We therefore adopted a similar protocol to that described by Sackton et al., whereby MPD was 'washed' out from the crystal prior to performing a soaking experiment with the desired ligand. We attempted these soaking experiments with our four highest affinity peptides, **D21**, **D20**, **D7**, and **D19** (in order of highest affinity to lowest) and were able to observe sufficient ligand density for all but **D19**.

The crystal structures of Cdc20$^{WD40}$ in complex with each of the other three D-box peptides (*Figure 5A–C*, *Figure 5—figure supplement 1*) show that they bound to Cdc20$^{WD40}$ at the canonical D-box degron binding site, with a largely similar topology to the *S. cerevisiae* Acm1-Cdh1 structure (*Figure 5D*; overlay of **D21** with Acm1 D-box). The guanidino group of the P1 arginine residue of the D-box peptides forms hydrogen bonds with the carboxylate side chains of Asp177 and Glu465 of Cdc20$^{WD40}$. The nitrogen backbone atom of the (*S*)–2-amino-4,4,4-trifluorobutanoic acid/(*S*)–2-amino-4,4-dimethylpentanoic acid unnatural amino acids also forms a hydrogen bond with the carbonyl of Asp177. Additionally, the carbonyl of S5 belonging to **D21**/**D20** forms a H-bond with the nitrogen backbone atom of Asp177. Lastly, Asp6 forms inter-molecular H-bonds with Arg174. We also observed an intra-molecular H-bond between the carbonyl of A2 with the amine of Gly5/Ser5, in addition to the carbonyl of A2 to the hydroxyl of Ser5 in **D21**/**D20**. Crystal packing of an adjacent asymmetric unit of the WD40 domain likely occludes the assumed binding site for the C-terminal three residues (…**ITN**-NH₂). We therefore conclude that this is the reason for the lack of observed density in this region of the peptides **D20** and **D21** (*Figure 5—figure supplement 1E and F*, respectively). We believe that it causes a reduction in binding affinities of all peptides *in crystallo*, given the evidence from SPR highlighting a role of position 7 in the interaction (*Table 1*). Interestingly, the observed electron density of the peptide correlates with Cdc20 binding affinity: **D21** and **D20**, having the highest affinities, display the clearest electron density allowing six amino acids to be modelled, whereas **D7** shows relatively poor density permitting modelling of only four residues. For **D19**, the lack of density observed

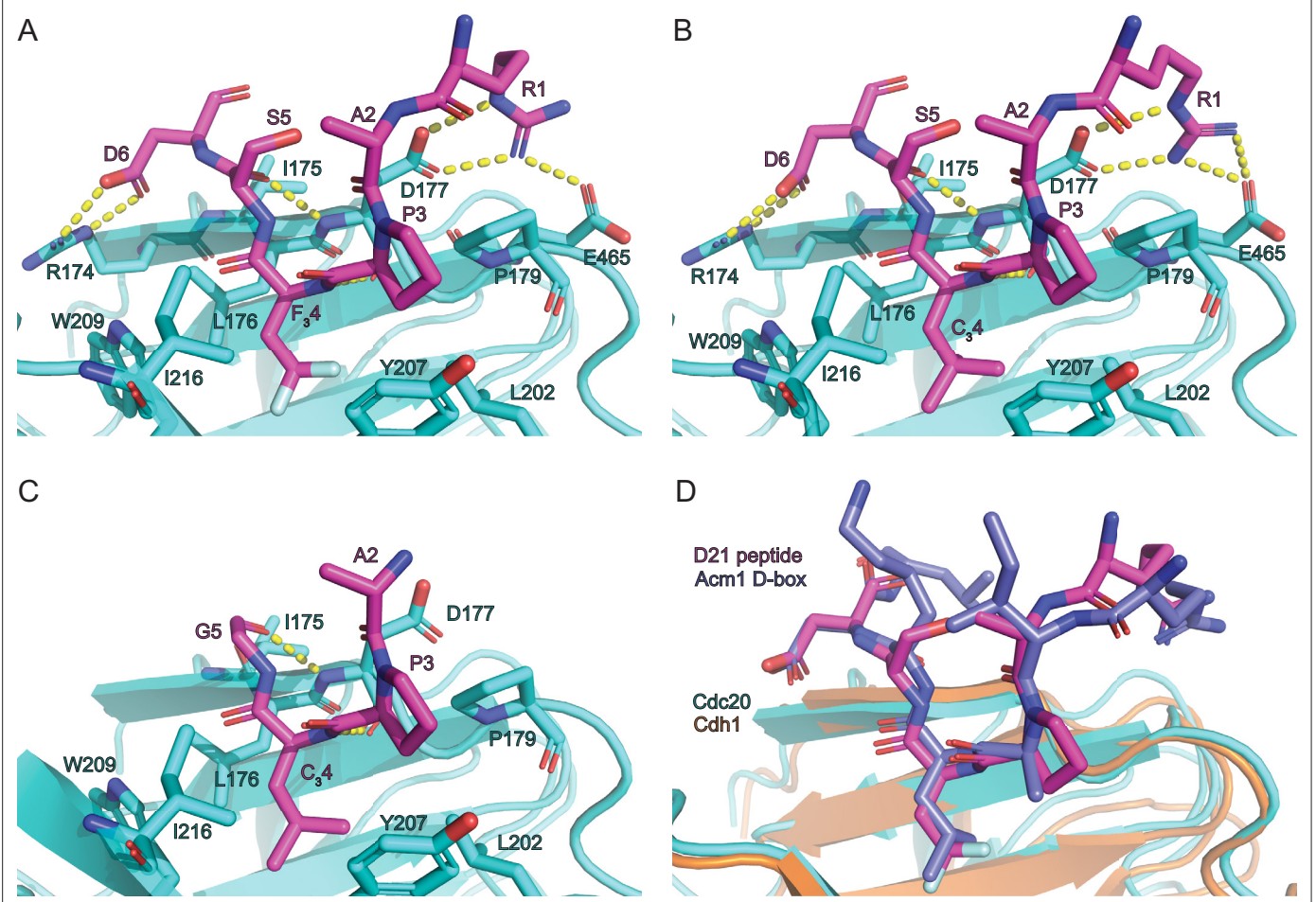

**Figure 5.** Crystal structures of Cdc20-D-box complexes. X-ray crystal structures of peptides (**A**) D21, (**B**) D20, and (**C**) D7 bound to the canonical D-box binding pocket of Cdc20. Intermolecular hydrogen bonds between peptides and Cdc20 are shown by dashed lines. (**D**) Structural alignment of D21-bound Cdc20 and Acm1 D-box peptide bound to Cdh1 (PDB: 4BH6; *He et al., 2013*). Peptide backbones align with an RMSD of 1.007 Å. Modelled water molecules have been removed from images for clarity.

The online version of this article includes the following figure supplement(s) for figure 5:

**Figure supplement 1.** Cdc20$^{WD40}$ is shown in cartoon (cyan) and surface (grey) representations.

likely reflects its intrinsically weaker affinity compared to the other peptides, in addition to losing the interactions from position 7 due to crystal packing. This hypothesis also correlates to the comments made by Tian et al. in their attempt to co-crystallise securin D-box peptides with Cdc20, in the identical space group (*Tian et al., 2012*). A recent 2.9 Å cryo-EM structure of the APC/C$^{Cdh1.Emi1}$ complex showed that the aliphatic residue at P7 interacts with Val178 of Cdh1 and would be predicted to contact the conserved Ile175 of Cdc20 (*Höfler et al., 2024*). This non-polar interaction between coactivator and D-box explains our observation that modifying the size of the aliphatic residue at P7 optimises D-box-Cdc20 affinity. Residues P8 and P9 of the D-box do not contact the coactivator but instead interact with APC10.

## D-box peptides bind to Cdc20 in the cellular context

We next investigated whether the four highest affinity peptides **D21**, **D20**, **D7**, and **D19** can bind to Cdc20 in the cellular context using CETSA (*Martinez Molina et al., 2013*). Sackton et al. previously demonstrated that Apcin can stabilise endogenous Cdc20 by using an isothermal CETSA method (*Sackton et al., 2014*). We were able to reproduce this ligand-induced stabilisation of Cdc20 with the more commonly used temperature gradient approach by densitometric analysis of western blots (*Figure 6—figure supplement 1A*). However, due to the low throughput of the assay we also explored

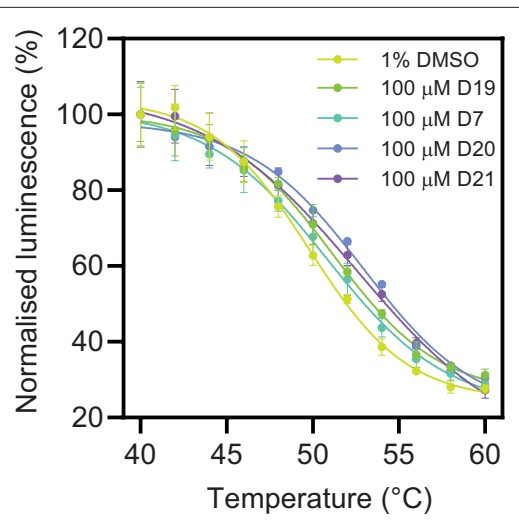

**Table 3.** Melting temperatures of HiBiT-tagged Cdc20 in the presence of 100 μM D-box peptides measured by cellular thermal shift assay.

Melting temperatures are calculated from the mean of three experiments, and the standard deviations are listed.

| Sample | Melting temperature (°C) |
|---|---|
| DMSO only | 50.0±0.4 |
| D19 | 50.4±0.6 |
| D7 | 51.2±0.3 |
| D20 | 52.6±0.4 |
| D21 | 53.2±0.8 |

**Figure 6.** D-box peptides bind to full-length HiBiT-tagged Cdc20 in the cellular context. Representative cellular thermal shift assay (CETSA) data are shown for Cdc20-tranfected HEK293T cell lysates incubated with D-box peptides at a concentration of 100 μM.

The online version of this article includes the following source data and figure supplement(s) for figure 6:

**Source data 1.** CETSA melting temperature data for *Figure 6*; *Table 3*.

**Figure supplement 1.** CETSA method development and validation using Apcin as a positive control.

**Figure supplement 1—source data 1.** Original file from one experimental replicate for western blot analysis of a CETSA experiment displayed in *Figure 6—figure supplement 1A*.

**Figure supplement 1—source data 2.** Original western blots labelled from the source data 1 used in *Figure 6—figure supplement 1A*.

**Figure supplement 1—source data 3.** Original file from one experimental replicate for western blot analysis of a CETSA experiment displayed in *Figure 6—figure supplement 1A*.

**Figure supplement 1—source data 4.** Original western blots labelled from source data 3 used in *Figure 6—figure supplement 1A*.

a more high-throughput approach by making use of Promega's split-luciferase HiBiT tag appended to the C-terminus of full-length Cdc20 and based on protocols previously described by Martinez and co-workers (*Martinez et al., 2018*). Notably, the signal is more sensitive and has a larger range compared to a western blot, and it removes a significant time-consuming centrifugation step from the workflow. We first confirmed that omitting the centrifugation step did not significantly affect the observed $T_m$ of vehicle control samples (*Figure 6—figure supplement 1B*). To further validate that the transfected Cdc20 is functional, we probed binding of 100 μM Apcin, which gave a $T_m$ of 54.4°C±0.6°C (*Figure 6—figure supplement 1C*). We then explored whether the D-box peptides at a fixed concentration stabilise the Cdc20, and for **D7**, **D20,** and **D21** we observed increases in the thermal stability of Cdc20 that correlated with their binding affinities as previously determined (*Figure 6*, *Table 3*). The lowest-affinity peptide, **D19**, did not result in a significant thermal stabilisation of Cdc20.

## D-box peptides inhibit APC/C$^{Cdc20}$ ubiquitination activity

We next assessed whether **D21** and **D20**, the two highest affinity peptides, are able to inhibit APC/C$^{Cdc20}$ activity. In the context of Cyclin B1 ubiquitination, we found that both peptides are more potent inhibitors compared with Apcin at the same concentration despite having slightly lower Cdc20-binding affinities than Apcin (*Figure 7*). However, in the context of the APC/C$^{Cdc20}$ complex, the D-box peptides, but not Apcin, would also contact APC10, thereby increasing their affinity for the APC/C-coactivator complex and explaining their greater inhibitory potency relative to Apcin.

## D-box peptides are able to target mNeon for degradation

To probe the functionality of the D-box variants at the cellular level, we conducted live cell degradation assays using mNeon fusions containing those peptide sequences that contain only natural amino acids: D1, D2, D3, and D19 (*Figure 8A*). The D-box sequences were swapped into an RxxL

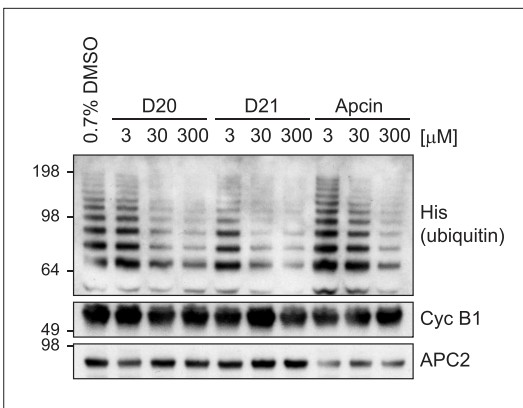

**Figure 7.** Inhibition of APC/C[Cdc20]-mediated ubiquitination of Cyclin B1 by D-box peptides and Apcin. In vitro ubiquitination assays using reconstituted APC/C[Cdc20] with Cyclin B1 as the substrate for ubiquitination. Lead peptides and Apcin were titrated from 3 µM to 300 µM and showed concentration-dependent inhibition of Cyclin B1 ubiquitination compared to the vehicle control (0.7% DMSO).

The online version of this article includes the following source data for figure 7:

**Source data 1.** Original western blots used in *Figure 7*.

**Source data 2.** Original western blots used in *Figure 7*.

motif previously shown to have no degron activity (**Abdelbaki et al., 2023**), which we here refer to as 'D0', adjacent to the endogenous C-terminal IDR of AURKA to enable processing of the ubiquitinated fusion proteins at the 26S proteasome. We found that all four new D-box variants tested could target mNeon for degradation, with timing consistent with targeting by APC/C[Cdc20] (**Figure 8B**). We predicted that the higher affinity D-box peptides from the in vitro assays (D1 and D19) would mediate increased rates and extent of degradation compared to the lower affinity peptides (D2 and D3). However, we found the opposite effect: D2 and D3 showed increased rates of mNeon degradation compared to D1 and D19 (**Figure 8C and D**). This observation is consistent with conclusions from other studies that affinity of degron binding does not necessarily correlate with efficiency of degradation. Indeed, there is no evidence that Hsl1, which is the highest affinity natural D-box (D1) used in our study, is degraded any more rapidly than other substrates of APC/C in yeast mitosis. A number of studies of a yeast 'pseudo-substrate' inhibitor Acm1 have shown that mutation of the high-affinity D-box in Acm1 converts it from inhibitor to substrate (**Choi et al., 2008**; **Enquist-Newman et al., 2008**; **Burton et al., 2011**) through a mechanism that governs recruitment of APC10 (**Qin et al., 2019**). Our study does not consider the contribution of APC10 to binding of our peptides to APC/C[Cdc20] complex, but since there is strong cooperativity provided by this additional interaction (**Hartooni et al., 2022**) we propose this as the critical factor in determining the ability of the different peptides to mediate degradation of mNeon.

## Discussion

Here, we quantified D-box peptide binding to Cdc20 and show that binding affinities can be enhanced by incorporating unnatural amino acids to better fill the hydrophobic pocket on the Cdc20 surface. We confirmed the success of this approach by determining X-ray crystal structures of Cdc20-peptide complexes. We showed target engagement by the peptides in the cellular context, and we found that the two highest affinity peptides were more potent inhibitors of APC/C[Cdc20] activity than the small molecule Apcin. Lastly, we found that the D-box peptide is a portable motif that can drive productive ubiquitination, leading to degradation when fused to a fluorescent protein target.

The finding that the peptides were more potent than Apcin as APC/C[Cdc20] inhibitors was interesting since Apcin has a slightly higher Cdc20-binding affinity than the peptides. It suggests that inhibiting APC/C[Cdc20] ubiquitination activity may require larger molecules to compete with substrates effectively. It may also be that, unlike Apcin, the peptides not only block the interaction of substrates with Cdc20 but additionally the interaction with APC10 and/or prevent the conformational change in APC/C that enables recruitment of the E2. In addition, although the inhibitory activity of the D-box peptides roughly correlates with their binding affinity, binding and degron activity (substrate degradation) are not directly correlated. Studies have shown that residues C-terminal of the D-box sequence in Acm1, the 'D-Box Extension' DBE motif, have the potential to inhibit reaction processivity by influencing the conformation of APC10 (**Qin et al., 2019**). The mNeon-D-box constructs used in our current study all

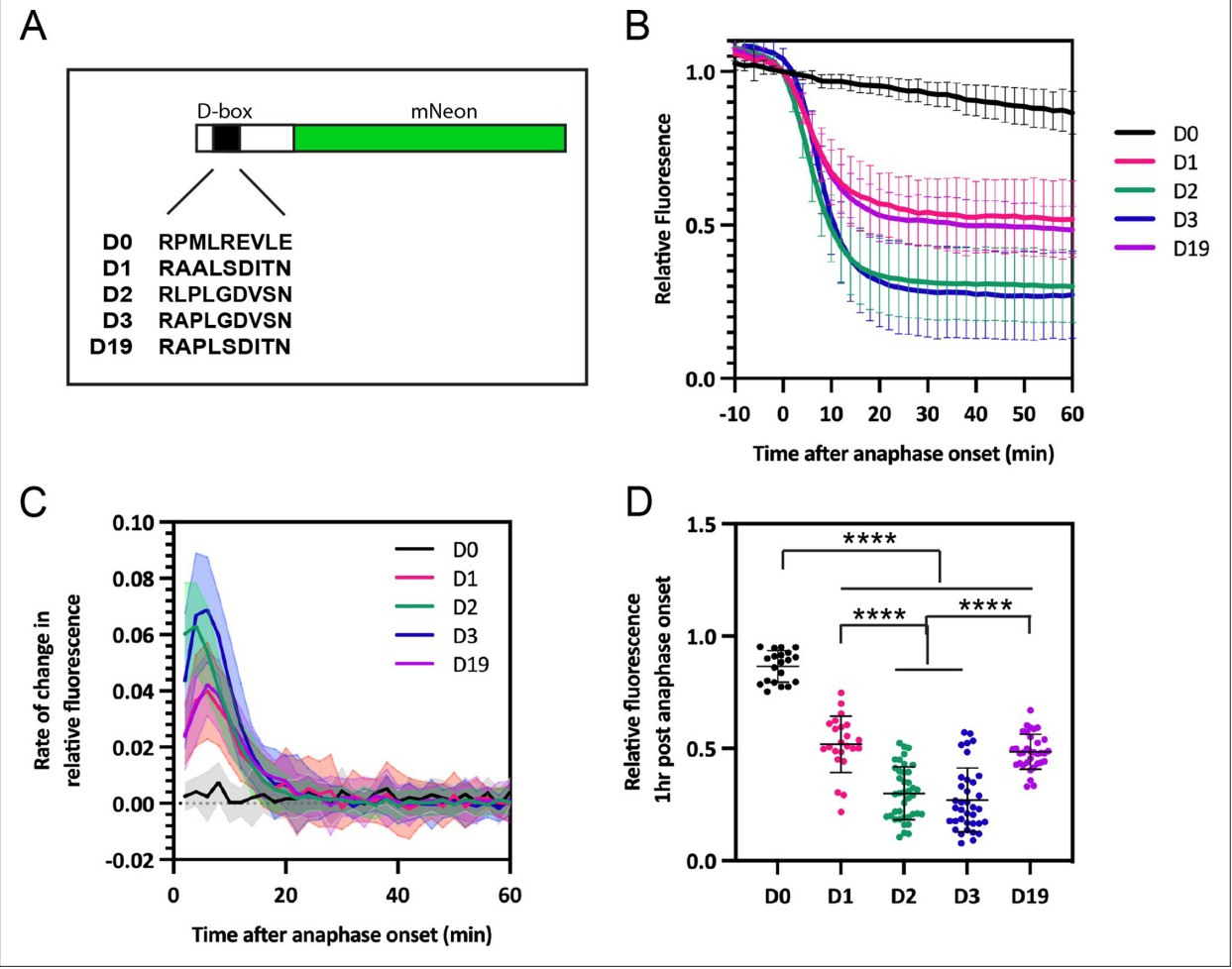

**Figure 8.** D-box variants can drive degradation in mitotic cells. (**A**) Schematic of D-box-mNeon constructs used in fluorescence timelapse imaging. (**B**) mNeon fluorescence levels in individual cells plotted over time to show D-box mediated degradation of mNeon in mitosis. Fluorescence measurements from individual cells are normalised to fluorescence at metaphase then in silico synchronised to anaphase onset. Mean degradation curves are shown, with error bars representing SDs. (**C**) Degradation rate curves show rate of change in relative fluorescence of the D-box variants and reveal maximum degradation rate for each construct. Error bars are depicted as shaded regions and indicate SDs. (**D**) Levels of relative fluorescence in each cell at t=1 hour after anaphase onset. Degradation of each D-box construct was significant relative to D0 control, using Welch's *t*-test. ****p≤0.0001. In (**B**–**D**), n=D0 (20) D1 (23), D2 (40), D3 (38), and D19 (34), with data pooled from two or more independent experiments.

The online version of this article includes the following source data for figure 8:

**Source data 1.** Data for the time dependence of mNeon fluorescence for *Figure 8*.

contain the same DBE motif, so a potential contribution from this motif will not affect the interpretation of our results, but it could certainly be added as an element in future inhibitor design.

In summary, the findings presented here represent a useful starting point for the further development of APC/C inhibitors as both research tools and also molecular therapeutics. Future directions could involve enhancing potency through avidity by incorporating multiple degrons into our molecules and additions to the D-box core sequence to include motifs that engage other components of the APC/C machinery – namely APC10 and the E2 – thereby not only blocking substrate binding more effectively but also better impeding ubiquitination activity. The results also have implications for the design of small-molecule and peptide-based degraders that harness the APC/C.

## Statistical analyses

GraphPad Prism 9 (GraphPad Software Inc) and Microsoft Excel (Microsoft Corporation) were used to analyse data, generate graphs, and perform statistical analyses. Statistical parameters, including the

sample size, the statistical test used, statistical significance (p-value), and the number of biological replicates, are reported in the figure legends or in the 'Materials and methods'.

## Materials availability

Materials from this study are available from the corresponding author upon reasonable request.

# Acknowledgements

We acknowledge funding of an AstraZeneca PhD studentship to RE, and Gates Cambridge Trust and Rosetrees Trust PhD studentships to CO. CL acknowledges funding from BBSRC grant BB/R004137/1 to her lab. DB acknowledges funding from MRC grant MC_UP_1201/6 and CRUK grant C576/A14109 to his lab. LSI acknowledges funding from the following to her lab: HFSP grant RGP0027/2020, CRUK grant C17838/A27225, and a Pancreatic Cancer Research Foundation Project grant. We thank Ziguo Zhang for providing the gene encoding Cdc20 in pU1, PIR1, and DH10-multibac$^{cre}$ cells. We thank David Fischer, Elizabeth Underwood, and Ross Overman for help with insect cell expression and purification of Cdc20, and Jason Breed for help with crystallographic data collection.

# Additional information

### Competing interests

Christopher Stubbs, Marianne Schimpl, Eileen J Fisher, Christopher Phillips: affiliated with Astra-Zeneca. The authors have no other competing interests to declare. The other authors declare that no competing interests exist.

### Funding

| Funder | Grant reference number | Author |
|---|---|---|
| Cancer Research UK | C17838/A27225 | Laura S Itzhaki |
| Pancreatic Cancer Research Fund | | Laura S Itzhaki |
| Human Frontier Science Program | 10.52044/hfsp. rgp00272020.pc.gr.165420 | Laura S Itzhaki |
| Biotechnology and Biological Sciences Research Council | BB/R004137/1 | Catherine Lindon |
| AstraZeneca | | Rohan Eapen |
| Rosetrees Trust | | Cynthia Okoye |
| Gates Cambridge Trust | | Cynthia Okoye |
| Medical Research Council | | David Barford |

The funders had no role in study design, data collection and interpretation, or the decision to submit the work for publication.

### Author contributions

Rohan Eapen, Conceptualization, Data curation, Formal analysis, Investigation, Writing – original draft, Writing – review and editing; Cynthia Okoye, Maria Zacharopoulou, Investigation, Writing – review and editing; Christopher Stubbs, Marianne Schimpl, Data curation, Formal analysis, Supervision, Investigation, Methodology, Writing – review and editing; Thomas Tischer, Formal analysis, Investigation, Writing – review and editing; Eileen J Fisher, Formal analysis, Investigation, Methodology, Writing – review and editing; Fernando Ferrer, Resources, Supervision, Investigation, Writing – review and editing, Methodology; David Barford, Resources, Formal analysis, Supervision, Methodology, Writing – review and editing; David R Spring, Supervision, Methodology, Writing – review and editing; Catherine Lindon, Resources, Data curation, Formal analysis, Supervision, Methodology, Writing – review and editing; Christopher Phillips, Conceptualization, Resources, Formal analysis, Supervision, Funding

acquisition, Project administration, Writing – review and editing; Laura S Itzhaki, Conceptualization, Resources, Formal analysis, Supervision, Funding acquisition, Writing – original draft, Project administration, Writing – review and editing

## Author ORCIDs
Rohan Eapen ⓘ https://orcid.org/0000-0001-7269-3633
Christopher Stubbs ⓘ https://orcid.org/0000-0002-2373-5478
Maria Zacharopoulou ⓘ https://orcid.org/0000-0002-3660-2797
David Barford ⓘ https://orcid.org/0000-0001-8810-950X
David R Spring ⓘ https://orcid.org/0000-0001-7355-2824
Catherine Lindon ⓘ https://orcid.org/0000-0003-3554-2574
Laura S Itzhaki ⓘ https://orcid.org/0000-0001-6504-2576

Reviewer #1 (Public review): https://doi.org/10.7554/eLife.104238.3.sa1
Reviewer #3 (Public review): https://doi.org/10.7554/eLife.104238.3.sa2
Author response https://doi.org/10.7554/eLife.104238.3.sa3

# Additional files

## Supplementary files
Supplementary file 1. Data collection, phasing, and refinement statistics for Cdc20 crystal structures with D-box peptides.

MDAR checklist

## Data availability
All data generated or analysed during this study are included in the manuscript and supporting files; source data files have been provided for Figures 2, 3, 4, 6, 7 and 8, and Figure 6-figure supplement 5. Diffraction data have been deposited in PDB under the accession codes 9I68, 9I69, 9I6A.

The following datasets were generated:

| Author(s) | Year | Dataset title | Dataset URL | Database and Identifier |
|---|---|---|---|---|
| Eapen R, Okoye C, Stubbs C, Schimpl M, Tischer T, Fisher EJ, Zacharopoulou M, Ferrer F, Barford D, Spring D, Lindon C, Phillips C, Itzhaki LS | 2025 | Crystal structure of human Cdc20 bound to synthetic D-box peptide D7 | https://doi.org/10.2210/pdb9i6a/pdb | Worldwide Protein Data Bank, 10.2210/pdb9i6a/pdb |
| Eapen R, Okoye C, Stubbs C, Schimpl M, Tischer T, Fisher EJ, Zacharopoulou M, Ferrer F, Barford D, Spring D, Lindon C, Phillips C, Itzhaki LS | 2025 | Crystal structure of human Cdc20 bound to synthetic D-box peptide D20 | https://doi.org/10.2210/pdb9i69/pdb | Worldwide Protein Data Bank, 10.2210/pdb9i69/pdb |
| Eapen R, Okoye C, Stubbs C, Schimpl M, Tischer T, Fisher EJ, Zacharopoulou M, Ferrer F, Barford D, Spring D, Lindon C, Phillips C, Itzhaki LS | 2025 | Crystal structure of human Cdc20 bound to synthetic D-box peptide D21 | https://doi.org/10.2210/pdb9i68/pdb | Worldwide Protein Data Bank, 10.2210/pdb9i68/pdb |

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
