## [Editor Report · eLife Assessment]

The article represents a **fundamental** advance in designing peptide inhibitors targeting Cdc20, a key activator and substrate-recognition subunit of the APC/C ubiquitin ligase. Supported by **compelling** biophysical and cellular evidence, the study lays a strong foundation for future developments in degron-based therapeutics. The revised article has been strengthened by additional clarifications and data that address prior reviewer concerns. The work provides a robust framework for developing tools to manipulate protein degradation and will be of broad interest to researchers in protein engineering, cell cycle regulation, and targeted protein degradation.

---

## [Referee Report · Reviewer #1 (Public review)]

Summary:

In this manuscript, the authors Eapen, et al. investigated the peptide inhibitors of Cdc20. They applied a rational design approach, substituting residues found in the D-box consensus sequences to better align the peptides with the Cdc20-degron interface. In the process, the authors designed and tested a series of more potent binders, including ones that contain unnatural amino acids, and verified binding modes by elucidating the Cdc-20-peptide structures. The authors further showed that these peptides can engage with Cdc20 in the cellular context, and can inhibit APC/CCdc20 ubiquitination activity. Finally, the authors demonstrated that these peptides could be used as portable degron motifs that drive the degradation of a fused fluorescent protein.

Strengths:

This manuscript is clear and straightforward to follow. The investigation of different peptide variations was comprehensive and well-executed. This work provided the groundwork for the development of peptide drug modalities to inhibit degradation or applying peptides as portable motifs to achieve targeted degradation. Both of which are impactful. The additional points provided by the authors in response to reviewers further strengthened the manuscript and enhanced its clarity.

Weaknesses:

None, the authors have addressed all my comments, and I have no additional suggestions.

---

## [Referee Report · Reviewer #3 (Public review)]

Summary:

Eapen and coworkers use a rational design approach to generate new peptide-inspired ligands at the D-box interface of cdc20. These new peptides serve as new starting points for blocking APC/C in the context of cancer, as well as manipulating APC/C for targeted protein degradation therapeutic approaches.

Strengths:

The characterization of new peptide-like ligands is generally solid and multifaceted, including binding assays, thermal stability enhancement in vitro and in cells, X-ray crystallography, and degradation assays.

Comments on revisions:

I am satisfied with the changes in response to the first round of review.

---

## [Author Response]

The following is the authors’ response to the original reviews

**Public reviews:**

**Reviewer #1 (Public review):**
Summary:In this manuscript, the authors Eapen et al. investigated the peptide inhibitors of Cdc20. They applied a rational design approach, substituting residues found in the D-box consensus sequences to better align the peptides with the Cdc20-degron interface. In the process, the authors designed and tested a series of more potent binders, including ones that contain unnatural amino acids, and verified binding modes by elucidating the Cdc-20-peptide structures. The authors further showed that these peptides can engage with Cdc20 in the cellular context, and can inhibit APC/CCdc20 ubiquitination activity. Finally, the authors demonstrated that these peptides could be used as portable degron motifs that drive the degradation of a fused fluorescent protein.Strengths:This manuscript is clear and straightforward to follow. The investigation of different peptide variations was comprehensive and well-executed. This work provided the groundwork for the development of peptide drug modalities to inhibit degradation or apply peptides as portable motifs to achieve targeted degradation. Both of which are impactful.Weaknesses:A few minor comments:(1) In my opinion, more attention to the solubility issue needs to be discussed and/or tested. On page 10, what is the solubility of D2 before a modification was made? The authors mentioned that position 2 is likely solvent exposed, it is not immediately clear to me why the mutation made was from one hydrophobic residue to another. What was the level of improvement in solubility? Are there any affinity data associated with the peptide that differ with D2 only at position 2?

The reviewer is correct that we have not done any detailed solubility characterisation; we refer only to observations rather than quantitative analysis. We wrote that we reverted from Leu to Ala due to solubility - we have clarified this statement (page 11) to say that that we reverted to Ala, as it was the residue present in D1, for which we observed a measurable affinity by SPR and saw a concentration-dependent response in the thermal shift analysis. We do not have any peptides or affinity data that explore single-site mutations with the parental peptide of D2. D2 is included in the paper because of its link to the consensus D-box sequence and thus was the logical path to the investigations into positions 3 and 7 that come later in the manuscript.

(2) I'm not entirely convinced that the D19 density not observed in the crystal structure was due to crystal packing. This peptide is peculiar as it also did not induce any thermal stabilization of Cdc20 in the cellular thermal shift assay. Perhaps the binding of this peptide could be investigated in more detail (i.e., NMR?) Or at least more explanation could be provided.

This section has been clarified (page 16). The lack of observed density was likely due to the relatively low affinity of D19 and also to the lack of binding of the three C-terminal residues in the crystal, and consequently it has a further reduced affinity. The current wording in the manuscript puts greater emphasis on this second aspect being a D19-specific issue, even though it applies to all four soaked peptides. The extent of peptide-induced thermal stabilisations observed by TSA and CETSA is different, with the latter experiment consistently showing smaller shifts. This observation may be due to the more complex medium (cell lysate vs. purified protein) and/or different concentrations of the proteins in solution. In the CETSA, we over-expressed a HiBiT-tagged Cdc20, which is present in addition to any endogenously expressed Cdc20. Although we did not investigate it, the near identical D-box binding sites on Cdc20 and Cdh1 would suggest that there will be cross-specificity, which could further influence the CETSA experiments.

The section now reads:

“We therefore assume that this is the reason for the lack of observed density in this region of the peptides D20 and D21 (Fig. S3E and S3F, respectively). We believe that it causes a reduction in binding affinities of all peptides in crystallo, given the evidence from SPR highlighting a role of position 7 in the interaction (Table 1). Interestingly, the observed electron density of the peptide correlates with Cdc20 binding affinity: **D21** and **D20**, having the highest affinities, display the clearest electron density allowing six amino acids to be modeled, whereas D7 shows relatively poor density permitting modelling of only four residues. For **D19**, the lack of density observed likely reflects its intrinsically weaker affinity compared to the other peptides, in addition to losing the interactions from position 7 due to crystal packing.”

**Reviewer #2 (Public review):**
Summary:The authors took a well-characterised (partly by them), important E3 ligase, in the anaphase-promoting complex, and decided to design peptide inhibitors for it based on one of the known interacting motifs (called D-box) from its substrates. They incorporate unnatural amino acids to better occupy the interaction site, improve the binding affinity, and lay foundations for future therapeutics - maybe combining their findings with additional target sites.Strengths:The paper is mostly strengths - a logical progression of experiments, very well explained and carried out to a high standard. The authors use a carefully chosen variety of techniques (including X-ray crystallography, multiple binding analyses, and ubiquitination assays) to verify their findings - and they impressively achieve their goals by honing in on tight-binders.Weaknesses:Some things are not explained fully and it would be useful to have some clarification. Why did the authors decide to model their inhibitors on the D-box motif and not the other two SLiMs that they describe?

For completeness, in addition to the D-box we did originally construct peptides based on the ABBA and KEN-box motifs, but they did not show any shift in melting temperature of cdc20 in the thermal shift assay whereas the D-box peptides did; consequently, we focused our efforts on the D-box peptides. Moreover, there is much evidence from the literature that points to the unique importance of the D-box motif in mediating productive interactions of substrates with the APC/C (i.e. those leading to polyubiquitination & degradation). One of the clearest examples is a study by Mark Hall’s lab (described in Qin et al. 2016), which tested the degradation of 15 substrates of yeast APC/C in strains carrying alleles of Cdh1 in which the docking sites for D-box, KEN or ABBA were mutated. They observed that whereas degradation of all 15 substrates depended on D-box binding, only a subset required the KEN binding site on Cdh1 and only one required the ABBA binding site. A more recent study from David Morgan’s lab (Hartooni et al. 2022) looking at binding affinities of different degron peptides concluded that KEN motif has very low affinity for Cdc20 and is unlikely to mediate degradation of APC/C-Cdc20 substrates. Engagement of substrate with the D-box receptor is therefore the most critical event mediating APC/C activity and the interaction that needs to be blocked for most effective inhibition of substrate degradation.

We have added the following text to the Results section “Design of D-box peptides” (page 10):

“We focused on D-box peptides, as there is much evidence from the literature that points to the unique importance of the D-box motif in mediating productive interactions of substrates with the APC/C (i.e. those leading to polyubiquitination & degradation). One of the clearest examples is a study that tested the degradation of 15 substrates of yeast APC/C in strains carrying alleles of Cdh1 in which the docking sites for D-box, KEN or ABBA were mutated ((Qin et al. 2017)). They observed that, whereas degradation of all 15 substrates depended on D-box binding, only a subset required the KEN binding site on Cdh1 and only one required the ABBA binding site. A more recent study (Hartooni et al. 2022) of binding affinities of different degron peptides concluded that KEN motif has very low affinity for Cdc20 and is unlikely to mediate degradation of APC/C-Cdc20 substrates. Engagement of substrate with the D-box receptor is therefore the most critical event mediating APC/C activity and the interaction that needs to be blocked for most effective inhibition of substrate degradation.”

What exactly do they mean when they say their 'observation is consistent with the idea that high-affinity binding at degron binding sites on APC/C, such as in the case of the yeast 'pseudo-substrate' inhibitor Acm1, acts to impede polyubiquitination of the bound protein'? It's an interesting thing to think about, and probably the paper they cite explains it more but I would like to know without having to find that other paper.

Interesting results from a number of labs (Choi et al. 2008, Enquist-Newman et al. 2008, Burton et al. 2011, Qin et al. 2019) have shown that mutation of degron SLiMs in Acm1 that weaken interaction with the APC/C have the unexpected consequence of converting Acm1 from APC/C inhibitor to APC/C substrate. A necessary conclusion of these studies is that the outcome of degron binding (i.e. whether the binder functions as substrate or inhibitor) depends on factors other than D-box affinity and that D-box affinity can counteract them. One idea is that if a binder interacts too tightly, this removes some flexibility required for the polyubiquitination process. The most recent study on this question (Qin et al.2019) specifically pins the explanation for the inhibitory function of the high affinity D-box in Acm1 on its ‘D-box Extension’ (i.e. residues 8-12) preventing interaction with APC10. In our current study, the binding affinity of peptides is measured against Cdc20. In cellular assays however, the D-box must also engage APC10 for degradation to occur. It may be that the peptide binding most strongly to the D-box pocket on Cdc20 is less able to bind to APC10 and therefore less effective in triggering APC10-dependent steps in the polyubiquitination pathway. The important Hartooni et al. paper from David Morgan’s lab confirms that even though the binding of D-box residues to APC10 is very weak on its own, it can contribute 100X increase in affinity of a peptide by adding cooperativity to the interaction of D-box with co-activator. Re Figure 6 and the fact that we did look at peptide binding in cells, these experiments were done in unsynchronised cells, so most Cdc20 would not be bound to APC/C.

We have modified the text (page 18) from:

“However, we found the opposite effect: D2 and D3 showed increased rates of mNeon degradation compared to D1 and D19 (Fig. 8C,D). This observation is consistent with the idea that high-affinity binding at degron binding sites on APC/C, such as in the case of the yeast ‘pseudo-substrate’ inhibitor Acm1, acts to impede polyubiquitination of the bound protein (Qin et al. 2019). Indeed, there is no evidence that Hsl1, which is the highest affinity natural D-box (D1) used in our study, is degraded any more rapidly than other substrates of APC/C in yeast mitosis. As shown in Qin et al., mutation of the high affinity D-box in Acm1 converts it from inhibitor to substrate (Qin et al. 2019). Overall, our results support the conclusions that all the D-box peptides engage productively with the APC/C and that the highest affinity interactors act as inhibitors rather than functional degrons of APC/C.”

to:

“However, we found the opposite effect: D2 and D3 showed increased rates of mNeon degradation compared to D1 and D19 (Fig. 8C,D). This observation is consistent with conclusions from other studies that affinity of degron binding does not necessarily correlate with efficiency of degradation. Indeed, there is no evidence that Hsl1, which is the highest affinity natural D-box (D1) used in our study, is degraded any more rapidly than other substrates of APC/C in yeast mitosis. A number of studies of a yeast ‘pseudo-substrate’ inhibitor Acm1, have shown that mutation of the high affinity D-box in Acm1 converts it from inhibitor to substrate (Choi et al. 2008, Enquist-Newman et al. 2008, Burton et al. 2011) through a mechanism that governs recruitment of APC10 (Qin et al. 2019). Our study does not consider the contribution of APC10 to binding of our peptides to APC/C^Cdc20^ complex, but since there is strong cooperativity provided by this additional interaction (Hartooni et al. 2022) we propose this as the critical factor in determining the ability of the different peptides to mediate degradation of associated mNeon.”

**Reviewer #3 (Public review):**
Summary:Eapen and coworkers use a rational design approach to generate new peptide-inspired ligands at the D-box interface of cdc20. These new peptides serve as new starting points for blocking APC/C in the context of cancer, as well as manipulating APC/C for targeted protein degradation therapeutic approaches.Strengths:The characterization of new peptide-like ligands is generally solid and multifaceted, including binding assays, thermal stability enhancement in vitro and in cells, X-ray crystallography, and degradation assays.Weaknesses:One important finding of the study is that the strongest binders did not correlate with the fastest degradation in a cellular assay, but explanations for this behavior were not supported experimentally. Some minor issues regarding experimental replicates and details were also noted.

Interesting results from a number of labs (Choi et al. 2008, Enquist-Newman et al. 2008, Burton et al. 2011, Qin et al. 2019) have shown that mutation of degron SLiMs in Acm1 that weaken interaction with the APC/C have the unexpected consequence of converting Acm1 from APC/C inhibitor to APC/C substrate. A necessary conclusion of these studies is that the outcome of degron binding (i.e. whether the binder functions as substrate or inhibitor) depends on factors other than D-box affinity and that D-box affinity can counteract them. One idea is that if a binder interacts too tightly, this removes some flexibility required for the polyubiquitination process. The most recent study on this question (Qin et al.2019) specifically pins the explanation for the inhibitory function of the high affinity D-box in Acm1 on its ‘D-box Extension’ (i.e. residues 8-12) preventing interaction with APC10. In our current study, the binding affinity of peptides is measured against Cdc20. In cellular assays however, the D-box must also engage APC10 for degradation to occur. It may be that the peptide binding most strongly to the D-box pocket on Cdc20 is less able to bind to APC10 and therefore less effective in triggering APC10-dependent steps in the polyubiquitination pathway. The important Hartooni et al. paper from David Morgan’s lab confirms that even though the binding of D-box residues to APC10 is very weak on its own, it can contribute 100X increase in affinity of a peptide by adding cooperativity to the interaction of D-box with co-activator. Re Figure 6 and the fact that we did look at peptide binding in cells, these experiments were done in unsynchronised cells, so most Cdc20 would not be bound to APC/C.

We have modified the text (page 18) from:

“However, we found the opposite effect: D2 and D3 showed increased rates of mNeon degradation compared to D1 and D19 (Fig. 8C,D). This observation is consistent with the idea that high-affinity binding at degron binding sites on APC/C, such as in the case of the yeast ‘pseudo-substrate’ inhibitor Acm1, acts to impede polyubiquitination of the bound protein (Qin et al. 2019). Indeed, there is no evidence that Hsl1, which is the highest affinity natural D-box (D1) used in our study, is degraded any more rapidly than other substrates of APC/C in yeast mitosis. As shown in Qin et al., mutation of the high affinity D-box in Acm1 converts it from inhibitor to substrate (Qin et al. 2019). Overall, our results support the conclusions that all the D-box peptides engage productively with the APC/C and that the highest affinity interactors act as inhibitors rather than functional degrons of APC/C.”

to:

“However, we found the opposite effect: D2 and D3 showed increased rates of mNeon degradation compared to D1 and D19 (Fig. 8C,D). This observation is consistent with conclusions from other studies that affinity of degron binding does not necessarily correlate with efficiency of degradation. Indeed, there is no evidence that Hsl1, which is the highest affinity natural D-box (D1) used in our study, is degraded any more rapidly than other substrates of APC/C in yeast mitosis. A number of studies of a yeast ‘pseudo-substrate’ inhibitor Acm1, have shown that mutation of the high affinity D-box in Acm1 converts it from inhibitor to substrate (Choi et al. 2008, Enquist-Newman et al. 2008, Burton et al. 2011) through a mechanism that governs recruitment of APC10 (Qin et al. 2019). Our study does not consider the contribution of APC10 to binding of our peptides to APC/C^Cdc20^ complex, but since there is strong cooperativity provided by this additional interaction (Hartooni et al. 2022) we propose this as the critical factor in determining the ability of the different peptides to mediate degradation of associated mNeon.”

**Recommendations for the authors:**

**Reviewer #1 (Recommendations for the authors):**
(1) On page 12 (towards the end), the author stated D10 contained an A3P mutation, they meant P3A right? 'To test this hypothesis, we proceeded to synthesise D10, a derivative of D4 containing an A3P single point mutation.'

We thank the reviewer for spotting this typo, which we have corrected.

(2) Have the authors considered other orthogonal approaches to cross-examine/validate binding affinities? That said, I do not think extra experiments are necessary.

We did not explore further orthogonal approaches due to the challenges of producing sufficient amounts of the Cdc20 protein. Due to the low affinities of many peptides for Cdc20, many techniques would have required more protein than we were able to produce. We believe that the qualitative TSA combined with the SPR is sufficient to convince the readers; indeed there is a correlation between SPR-determined binding affinities and the thermal shifts: For the natural amino acid-containing peptides (Table 1) D19 has the highest affinity and causes the largest thermal shift in the Cdc20 melting temperature, D10 has the lowest affinity and causes the smallest thermal shift, and D1, D3, D4, and D5 and all rank in the middle by both techniques. For those peptides containing unnatural amino acids (Table 2), again higher affinities are reflected in larger thermal shifts.

**Reviewer #2 (Recommendations for the authors):**
The data seem fine to me. I would appreciate a little more detail on the points mentioned in the public review. Also a thorough reread, maybe by a disinterested party as there are various typos that could be corrected - all in all an excellent clear paper that encompasses a lot of work.

A colleague has carefully checked the manuscript, and typos have been corrected.